# Medial Calcar Comminution and Intramedullary Nail Failure in Unstable Geriatric Trochanteric Hip Fractures

**DOI:** 10.3390/medicina57040338

**Published:** 2021-04-01

**Authors:** Seth M. Tarrant, David Graan, Drew J. Tarrant, Raymond G. Kim, Zsolt J. Balogh

**Affiliations:** 1Department of Traumatology, John Hunter Hospital, Lookout Rd, New Lambton Heights, NSW 2305, Australia; seth.tarrant@uon.edu.au (S.M.T.); David.Graan@hnehealth.nsw.gov.au (D.G.); drew.tarrant@health.nsw.gov.au (D.J.T.); Raymond.Kim@health.nsw.gov.au (R.G.K.); 2School of Medicine and Public Health, University of Newcastle, Callaghan, NSW 2308, Australia

**Keywords:** hip fracture, osteoporosis, geriatric, fragility, fracture, femoral nail, trochanteric, fixation, failure

## Abstract

*Background and Objectives*: An increasing global burden of geriatric hip fractures is anticipated. The appropriate treatment for fractures is of ongoing interest and becoming more relevant with an aging population and finite health resources. Trochanteric fractures constitute approximately half of all hip fractures with the medial calcar critical to fracture stability. In the management of unstable trochanteric fractures, it is assumed that intramedullary nails and longer implants will lead to less failure. However, the lack of power, inclusion of older generation femoral nails, and a variable definition of stability complicate interpretation of the literature. *Materials and Methods*: Between January 2012 and December 2017, a retrospective analysis of operatively treated geriatric trochanteric hip fracture patients were examined at a Level 1 Trauma Centre. The treatment was with a long and short version of one type of trochanteric nail. Unstable trochanteric fractures with medial calcar comminution were examined (AO31A2.3, 2.3 & 3.3). The length of the medial calcar loss, nail length, demographics, fracture morphology, and relevant technical factors were examined in univariate and multivariate analysis using competing risk regression analysis. The primary outcome was failure of fixation with post-operative death the competing event and powered to previously reported failure rates. *Results*: Unstable patterns with medial calcar comminution loss constituted 617 (56%) of operatively treated trochanteric fractures. Failure occurred in 16 (2.6%) at a median post-operative time of 111 days (40–413). In univariate and multivariate analysis, only younger age was a significant predictor of failure (years; SHR: 0.91, CI 95%: 0.86–0.96, *p* < 0.001). Nail length, medial calcar loss, varus reduction, and other technical factors did not influence nail failure. *Conclusions*: In a cohort of unstable geriatric trochanteric hip fractures with medial calcar insufficiency, only younger patient age was predictive of nail failure. Neither the length of the medial calcar fragment or nail was predictive of failure.

## 1. Introduction

There will be an increasing global burden of geriatric hip fractures in the future. Extracapsular, or trochanteric, proximal femur fractures constitute approximately 50% of hip fracture types, with variations evident in national registries [1]. Intramedullary nailing (IMN) is an increasingly popular fixation method for trochanteric fractures, despite concerns over cost and increased mortality [2]. The better reliability of IMN compared to extramedullary constructs in ‘unstable’ fracture variants [3] has led to its preference in complex fractures [4].

‘Stability’ itself has been attempted to be defined. The trochanteric fracture variants of the AO/OTA classification (Arbeitsgemeinschaft für Osteosynthesefragen/Orthopedic Trauma Association) have been recently revised to reflect the importance of lateral wall insufficiency in defining stability [5]. Knobe et al. established in a nationwide German survey of orthopaedic trauma surgeons that ‘absence of medial support’ is the most critical component for determining fracture stability [6]. Femoral nails have shown to have lower fatigue limits in fracture patterns with posteromedial comminution and support loss of the lesser trochanter (LT) [7]. In vitro experimentation has demonstrated that an increase in LT fragment volume is associated with earlier nail failure [8] and increased medial buttress comminution is related to increased implant loading [9]. Clinically, larger posteromedial fragmentation leads to greater collapse of the neck as evidenced by lateralisation of IMN lag screws [10] and the presence of a posteromedial fragment (‘yes’ or ‘no’) is a predictor for lag screws cut-out in IMN [11].

How the nail construct, most notably nail length, affects failure in unstable fractures is not well defined. Previous studies investigating nail length and failure have likely been underpowered [12,13,14,15,16,17] or use a variety of nailing systems with some already antiquated [18,19]. For randomized studies, adequately powering to failure as an outcome has not been performed [20], or a mixture of stable and unstable fracture patterns have been recruited without subgroup analysis [21]. Large data sets and registries with adjustment for confounders may be able to demonstrate effectiveness in answering such questions. 

This study aims to utilise a large volume centre’s operative data to examine what variables, including nail length, affect the need for revision surgery in an adequately powered cohort of unstable trochanteric fractures. We hypothesise that nail length itself in a non-randomised context will not affect failure of fixation.

## 2. Materials and Methods

### 2.1. Patients and Settings

The study was conducted at a University affiliated Level 1 Trauma Centre, which is also a primary to tertiary referral centre for hip fractures and admits over 400 geriatric hip fractures per year [22,23]. All patients were identified through the prospectively collected institutional Long Bone Fracture Database and the Australia New Zealand Hip Fracture Registry (ANZHFR). The institutions operative database was cross-checked. Ethical approval was obtained from Hunter New England Health Ethics Committee (Ethics number HNEHREC: AU201903-08 approved on 8 March 2019)

#### 2.1.1. Inclusion 

Patients included were aged 65 years or older who sustained a low-energy hip fracture. Fractures were included if they were pertrochanteric and intertrochanteric proximal femur fractures and treated with a Gamma 3 (Stryker, Kalamazoo, MI, USA) nail. In concordance with expert opinion and biomechanical studies highlighting the importance of the LT, we focused on unstable patterns with a fractured and displaced LT. This equated to the AO/OTA subtypes AO31A2.2, 2.3 and 3.3 [5]. AO31A2 fractures involve the trochanteric region with multiple fragments (including the LT) and lateral wall incompetency. AO31A2.2 is defined by having 1 intermediary fragment (this fragment is between the neck and the diaphysis involving the greater trochanter (GT) whilst AO31A2.3 has multiple fragments in this region. AO31A3.3 is a multifragmentary pattern characterised by a fracture line extending from the medial calcar of the femoral neck to the lateral femoral cortex. The LT is detached as a separate fragment in this pattern [5].

#### 2.1.2. Exclusion

The exclusion criteria was high-energy mechanisms and polytrauma, pathological fractures (confirmed primary or metastatic lesion on intra-operative histopathology), peri-implant and periprosthetic fractures.

### 2.2. Data Collection

Demographic data was collected prospectively for registry patients (age, sex and American Society of Anesthesiologists (ASA) score, post-operative weight-bearing status). A retrospective review was conducted as needed for pre-ANZHFR patients (pre-February 2015). Comorbidities were collected from coding (ischaemic heart disease, chronic renal failure, diabetes mellitus, current and past tobacco use, body mass index 35 and above). Radiographs were assessed at least 1 year after index surgery by 3 orthopaedic trained doctors, and contention over classification was settled by a 4th reviewer. Picture Archiving and Communication systems (PACS) was accessed to view all radiographic imaging within the State public hospital sector and measurements made to digital scales. Fluoroscopic imaging was scaled to the diameter of the lag screw (10.5 mm). Mortality figures were derived from the ANZHFR and the New South Wales (NSW) Registry of Births, Deaths and Marriages.

Clinical follow up for geriatric hip fracture is rarely conducted past 6 weeks post-operatively in our institution due to the considerable logistical burden to often institutionalized and cognitively impaired patients. As emergency departments exclusively exist within the public health sector and upload images to a Statewide imaging system, an assumption is made that construct failures requiring revision surgery would be detected.

### 2.3. Outcomes

The primary outcome of failure was defined as an unexpected surgical revision. This did not include infection, or removal of hardware from irritation or exchange of lag screw lateral prominence. Failures included peri-implant fracture, non-union, screw cut out and implant failure in context of not united fracture. Non-union was not differentiated into atrophic or hypetrophic in keeping with contemporary theories of bone healing [24]. The competing risk for failure was post-operative death.

### 2.4. Technical Factors of Fixation

Surgery was conducted by orthopaedic surgeons or advanced orthopaedic surgical trainees under consultant supervision. The proximal engagement of the nail was defined as contact with the trochanteric cortex or protrusion. Lateral fixation, previously shown to be critical to fixation constructs [14], was defined by complete contact with the lateral femoral entry point. Only partial contact with the inferior cortex was not included as adequate lateral screw engagement of the femoral cortex [14]. The Cleveland index of the lag screw was based upon intraoperative images [25] where appropriate rotation and orthogonal views were achieved. These images were additionally used to derive both tip apex distance (TAD), calcar-reference TAD [26] and TADcalTAD [27]. Neck shaft angulation post-operatively of more than 5 degrees varus [28] from the uninjured side was documented as a critical marker of malreduction. In the instance of contralateral fracture or arthroplasty, previous radiographs were used. Medial calcar loss was measured as the craniocaudal (or superoinferior) LT fragment length on the preoperative digital image with PACS. In instance of comminution, intraoperative (fluoroscopic) calcar deficit upon restoration of femoral alignment was measured. Distal locking was classified as at least one bicortical screw through the end of the nail.

### 2.5. Sample Size

A power calculation with Pocock formula was conducted with an alpha value of 0.05 and beta value of 0.2 (power 80%) based upon the mean failure rates of short and long Gamma 3 IMN (introduced to market in 2003 and our centre in 2005) [12,14,15,29,30,31]. A minimum of 198 patients for both long and short nails were needed to detect a difference. The identification of patients was conducted in backwards basis per annum starting from 2017 to allow at least 1 year of radiographic follow up. 

### 2.6. Statistical Analysis

Continuous data were assessed for distribution and presented as mean and standard deviation, or median and interquartile ranges. Categorical data are presented as a count and percentage. 

The primary outcome of implant failure was assessed with competing risk-regression analysis due to the high mortality associated with geriatric hip fracture. The cumulative incidence of failure was estimated as a function of postoperative day (POD), with death as a competing event according to the Fine and Gray model. No events were censored with the assumption that this population would not relocate from the state, and hence both death and failure would be detected. Subdistribution hazard ratios (SHR; the probability of an event occurring over time) and confidence intervals are presented for interpretation. 

Univariate analysis was performed for demographic factors (age, sex, ASA as a surrogate marker of comorbidity) and technical factors (proximal engagement, lateral wall engagement, TAD, calTAD, TADcalTAD, distal locking, varus malreduction and femoral head screw position) that may affect failure. Independent variables with *p* < 0.2 were entered into a multivariate model. On a priori basis, nail length and the interaction between medial calcar fracture size length and fracture pattern were entered into the multivariate model. This was performed to account for selection bias that fractures with a longer medial calcar component would receive longer nails, and that increasing comminution is additionally biased towards longer nails. Statistical analyses were programmed using Stata v13.0 (StataCorp LP, College Station, TX, USA).

## 3. Results

### 3.1. Patient Selection & Demographics

To generate adequate groups 6 years of data was examined: January 2012–December 2017 (Appendix A). This resulted in 396 (64%) patients in the short nail group and 221 (36%) patients in the long nail group. Median length of radiographic follow up was 707 days (survival: 1–2634) with 371 (60%) deaths. Failed osteosynthesis occurred in 16 (2.6%) patients. Demographics, transfusion and mortality is tabulated in Appendix B. Individual patient details are tabulate in Appendix C. 

### 3.2. Nail Length & Medial Calcar Loss

The median calcar loss per fracture type was 48 mm (±19 mm) for A2.2, 48 mm (±19 mm) for A2.3 and 52 mm (±32 mm) for A3.3 patterns. Longer nails were used for patterns with an increased mean calcar loss for all fracture patterns (Table 1).

### 3.3. Modes of Failure

The median time to failure was 111 days (40–413) (Table 2). Post-operative fracture occurred in 7 (1.1%) patients (either peri-implant (Figure 1) or distal to implant) at a median of 182 days (40–450). The other failure modes in 9 (1.5%) patients (screw cut-out (Figure 2), implant breakage and non-union (Figure 3) occurred at 111 days (80–413).

### 3.4. Failure

Univariate predictors of failure are presented in Appendix D. The only significant predictor of failure was age, with decreasing years being associated with a higher probability of failure (year; SHR: 0.90, CI 95%: 0.86–0.95, *p* < 0.001). Predictors under *p* < 0.2 included age, TAD (mm; SHR: 1.06, CI 95%: 0.99–1.13, *p* = 0.102), TADcalTAD (mm; SHR: 1.03, CI 95%: 0.99–1.06, *p* = 0.149) and calcar loss (mm; SHR: 1.02, CI 95%: 1.00–1.04, *p* = 0.117). Unadjusted comparison of long vs short nails showed no difference in failure rate (long; SHR: 1.09, CI 95%: 1.60–1.04, *p* = 0.117) (Figure 4). The interaction between nail length and calcar loss showed a *p* < 0.2 for both short (mm; SHR: 1.02, CI 95%: 0.99–1.06, *p* = 0.162) and long nails (mm; SHR: 1.02, CI 95%: 0.99–1.04, *p* = 0.139). TADcalTAD was eliminated from the multivariate model due to its correlation with TAD (*r* = 0.92). As there were no failure events in the 31A3.3 group that received short nails (*n* = 7), nail length could not be appropriately compared for this pattern. The results from the multivariate model are in Table 3. Only age remained a significant predictor of failure in the multivariate model (years; SHR: 0.91, CI 95%: 0.86–0.96, *p* < 0.001) (Table 3).

## 4. Discussion

This study analysed 1110 consecutive geriatric hip fracture patients undergoing surgical fixation with a Gamma 3 (G3) IMN over a 6-year period. In a cohort of 617 patients with lateral wall insufficiency and medial calcar loss, variables including non-randomised nail length and the craniocaudal length of medical calcar loss were examined. In both univariate and multivariate survival analysis, only younger age was associated with failure. 

### 4.1. Rates of Failure 

This study identified 16 cases (2.6%) of failures This is comparable to Vaughan et al. [12], reporting a failure rate of 3.1% using G3 IMN. Other studies include total complication ranging between 5.4 and 13% [13,16,19]. This low failure rate is potentially due to the relatively high volume of nailing for hip fractures in our centre [23] and the use of the G3 IMN for 7 years before the first patient studied in this cohort. 

In assessing for patient, technical and radiographic factors contributing to failure, only younger age (years; SHR: 0.91, CI 95% 0.86–0.96, *p* < 0.001) was a statistically significant risk factor for failure. These findings may suggest that younger populations are more independent, potentially mobilise longer distances, hence cycle and stress the IMN construct and potentiate failure. A younger age has previously been correlated to intramedullary nail failure in a general population of proximal femur fracture patterns [28]. 

The overall secondary fracture rate was 1.1%. This rate correlates to existing studies reporting an overall post-operative secondary fracture rate of 0.5–4.2% with no difference demonstrated with nail length [12,13,15,19]. Median time to fracture was 182 days, later than the median of non-fracture modes. Horner et al. [19] reported a 4.2% fracture rate in the Gamma 1 and 2 systems. With a minimum 13-year follow-up and 56% of fractures occurring more than 1.5 years post index surgery. The limited median follow-up of 707 days (or 1.9 years) in this study may limit the true prevalence of fracture. 

### 4.2. Nail Length

In simple univariate analysis, there appeared to be no difference in failure type or rate between long and short nails (short; SHR: 1.09, CI 95% 0.40–3.00, *p* = 0.866). Small failure numbers limited the ability to adjust for bias in multivariate analysis, evidenced by large confidence intervals. 

Our centre demonstrated a disposition towards short nails contrasting to other series with long nails comprising the majority of implants [12,13,15]. Short nails have been shown to utilise approximately 20–30 min less operative time with less estimated blood loss in randomised trials, which is desirable for surgical efficiency and patient welfare [21]. Despite being a topic of recurrent interest, there is no proven association of increased risk of overall failure in comparison of long versus short IMN [12,13,15,16,17,19,21], with further metanalysis showing no difference in nail failure for length [32]. However, great variability in fracture patterns prevails with bias towards longer nails used for more complex fractures. Whilst the majority of these studies include both stable A1 and unstable fracture A2 patterns (often excluding the A3 patterns), the role of medial calcar is not investigated. 

### 4.3. Medial Calcar

The loss of the medial calcar through fracture fragmentation is considered critical for fracture stability [6]. The decision to exclude the 2-part A3.1 and A3.2 fractures was made to model medial calcar loss along with other univariate variables. Biomechanical studies demonstrate posteromedial comminution, LT loss and LT volume as a function of greater trochanter volume are associated with earlier nail failure [8]. We calculated the craniocaudal length of the medial calcar loss and adjusting for bias in nail length selection demonstrated it was not predictive of failure (short nail failure; SHR: 1.03, 95% CI 0.98–1.07, *p* = 0.204/long nail failure; SHR 0.98, CI 95% 0.93–1.04, *p* = 0.518). The restoration of the medial calcar during surgery was not assessed as an independent variable. Whilst fixation of a displaced LT has been proposed in order to reduce complications such as ‘thigh pain’ [33], this has not been reproduced elsewhere in the literature and is not associated with decreasing failure. Attempts at anatomic fixation add time to a procedure and invite complications without proven benefit. 

### 4.4. Limitations

Limitations included retrospective data collection relying on documentation in medical records and simple radiographic assessment by three analysers. We did not perform tests of inter—or intraobserver reliability for the measurements. Radiographs to judge stability with the AO classification has its limitations. Whilst computed tomography can more accurately define fracture morphology, and hence complexity, of trochanteric fractures, it is far from standard practice [34]. Hence, radiograph-based decision making will be the mainstay for the immediate future. With the burden of osteoporotic fractures increasing rapidly in low-and middle-income countries, access to three-dimensional imaging modalities is most likely unfeasible.

As a retrospective cohort study, the study may be subject to implant selection bias. Whilst attempting to account for this with adjusted analysis, the lower failure rates, particularly of short nails, than reported in the literature lead to statistical power loss. Based upon the simple SHR from the univariate analysis (SHR: 1.09), to appropriately power a study examining long vs short nails, it would require over 4000 patients. It is unlikely a single institution would be able to answer this question.

A medium to long-term clinical follow up of geriatric hip fracture patients is not routine. Representation to an acute facility with fixation failure relies on the assumption that elderly and non-migratory patients will remain within the State. Additional assumptions include that complications which necessitate revision will represent with pain or dysfunction, a limitation in a cohort with over 30% cognitive impairment and 27% living in an aged care facility [23]. With median failure occurring beyond 90 days (typically the final ‘3 month’ follow up timepoint in an orthopaedic fracture clinics), it is not apparent from this study whether more failures would have been recruited with routine follow up. 

The differences in fracture morphology are a further limitation. The reverse obliquity present in the A3.3 variant is perceived as a ‘more unstable’ pattern than A2 variants. Despite randomised evidence suggesting equivalence for long and short nails in this A3 fractures [20], there is lack of power and no subdivision or consideration of medial calcar loss. At our institution over 90% of for A3.3 fractures received long nails. With no failures using short nails (*n* = 7) there was no meaningful comparison statistically generated. Despite our centre using more short nails than many reports in the literature [12,13,15], the predisposition towards long nails in the A3.3 variant implies the distal extent of the lateral wall fracture may be of significance. This was not collected as an independent variable for model generation as it is not applicable for A2 patterns. 

Heterogeneity of the definition of ‘failure’ may also limit the application of this study. We did not consider femoral neck collapse and lateral screw prominence necessitating exchange as a failure. Removal of hardware has been reported in other studies and cited at 7.5%, with lag screw exchange less than 0.1% [35]. As there was no universal follow up until union, we could not assess the extent of femoral offset loss. This has been cited at a mean of 5.9 mm in unstable AO patterns [36]. Whilst related to functional impairments [36], the relation between controlled impaction and failure is not known, and not examined in this study. 

Whilst peri-implant fracture has may not been considered a technical failure in some studies, this has also been tied to the belief that shorter nails were predisposed to fracture by design and was unique to short nails [14]. The dogma that the choice of implant needs to ‘span the bone’ in unstable variants led to the inclusion of peri-implant fracture as a failure mode. If we demonstrated a higher failure or fracture rate with a short nail, this would certainly be a salient message. However surgical selection bias has not demonstrated this. 

Finally, whilst identifying that age was predictive of failure, this was not correlated with other variables such as pre-morbid mobility, cognition and discharge destination. Whilst age may be a surrogate marker of these factors, it is not known whether they would have influenced failure. 

### 4.5. Strengths

This study reviewed 1100 cases of hip fractures fixated with the G3 IMN, of which 617 unstable patterns with medial calcar loss were analysed. This is a comparatively large sample size compared to current studies, with the largest identified involving 644 patients and early generation femoral nails [19]. The strengths of this study include CRR analysis to assess the rate of failure while accounting for the competing risk of death [37] as Cox Regression may over-estimate failure rates [13,18,19]. Whilst mortality was not a focus of this paper, it is inherently analysed. Paucity of long-term mortality in the setting of failure challenges interpretation [21]. 

Despite anecdotal rules regarding the size of the medial calcar loss and subsequent choice of nail length, this has not been studied in vivo. Hence, we believe that this measurement is novel, and a potentially meaningful variable to analyse, particularly as awareness grows around the importance of three-point proximal fixation [14] and other technical factors [3,26].

The findings of our study hold external validity as both the G3 and IMN procedure is common practice, and our demographics are similar to global hip fracture populations [1]. Furthermore, our breakdown of 31A-type fractures is an almost identical distribution to other trauma centres [34].

## 5. Conclusions

In a cohort of geriatric trochanteric hip fractures with lateral wall and medial calcar insufficiency, younger patient age was predictive of nail failure. As a retrospective study, bias was attempted to be adjusted for with examination of multiple demographics, technical and fracture characteristics. However, clinical acumen driving selection bias likely persists. Neither the length of the nail nor medial calcar fragmentation predicted failure.

## Figures and Tables

**Figure 1 medicina-57-00338-f001:**
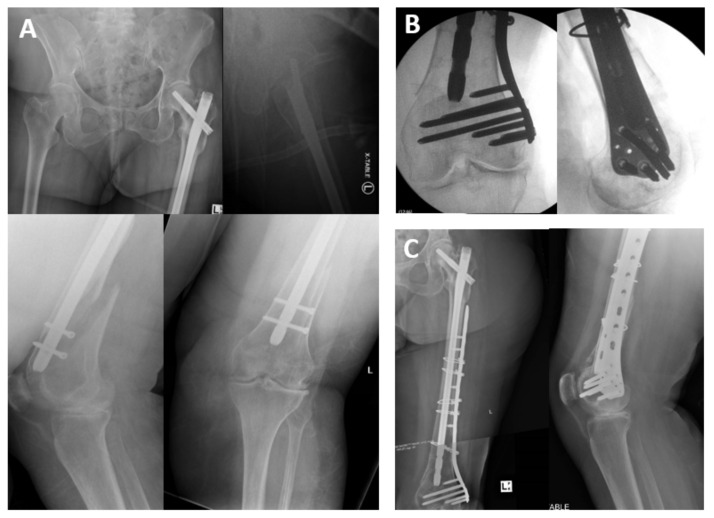
998 days post-treatment for A3.3 fracture, a peri-implant fracture around the distal nail is sustained (**A**). This is managed by removal of the distal screws and plate osteosynthesis (**B**) followed by uneventful fracture union (**C**).

**Figure 2 medicina-57-00338-f002:**
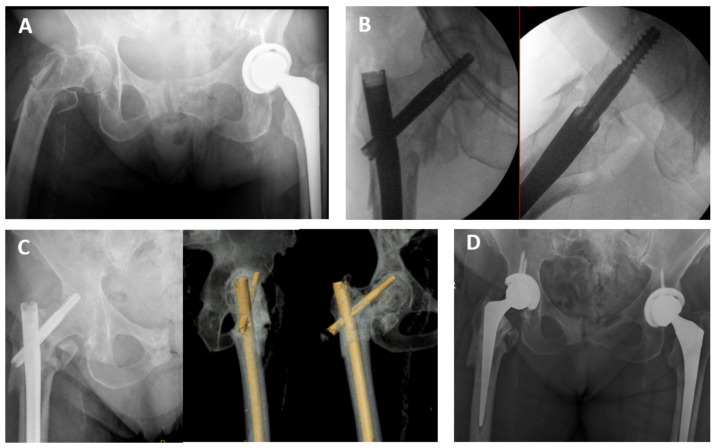
A comminuted A3.3 fracture variant with a 32 mm length lesser trochanter fragment treated with a proximally engaged long femoral nail post-inclusion period (**A**). The lag screw despite having an acceptable TAD of 18 mm and lateral engagement is in Cleveland zone 4 (**B**). It was cut out at post-operative day 287 (**C**) and was revised to a hybrid total hip replacement (**D**).

**Figure 3 medicina-57-00338-f003:**
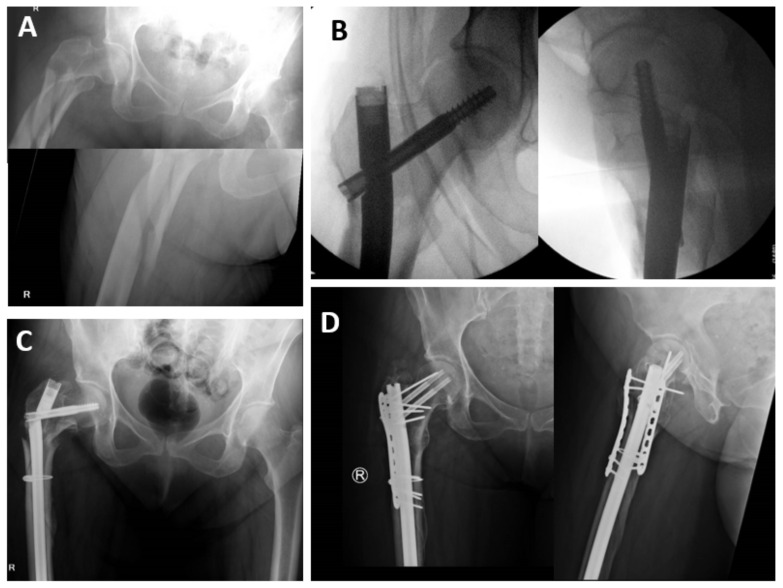
A 70-year-old female with a 136 mm loss of medial calcar continuity in a A3.3 pattern (**A**) has a technically sufficient proximal reduction with a long nail and single cable (**B**). The construct fails at post-operative day 81 (**C**) and is revised with reconstruction nail with dual plating (**D**).

**Figure 4 medicina-57-00338-f004:**
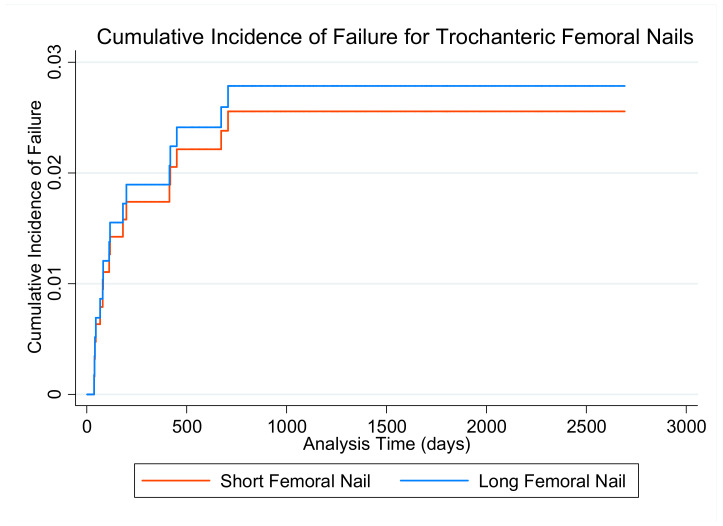
Cumulative incidence of failure for short and long trochanteric nails.

**Table 1 medicina-57-00338-t001:** Medial calcar loss and nail length.

Medial Calcar Loss (mm)	All Patients	Short Nail	Long Nail
	(*n* = 617)	(*n* = 396)	(*n* = 221)
All fracture types (*n* = 617)	48 (±22)	43 (±16)	57 (±27)
Subtypes			
2.2 (*n* = 452)	48 (±19)	44 (±16)	60 (±23)
2.3 (*n* = 77)	48 (±19)	40 (±17)	57 (±27)
3.3 (*n* = 88)	52 (±32)	33 (±9.0)	53 (±33)

**Table 2 medicina-57-00338-t002:** Failure mode.

Failure Mode	Total (*n* = 617)	Short (*n* = 396)	Long (*n* = 221)
Screw Cut-out	2 (0.2%)	2 (0.5%)	0 (0%)
Implant Breakage (Delayed Union)	3 (0.3%)	1 (0.3%)	2 (0.9%)
Fracture Distal to Implant	3 (0.3%)	3 (0.8%)	0 (0%)
Peri-implant Fracture	4 (0.6%)	1 (0.3%)	3 (1.3%)
Non-union	4 (0.6%)	3 (0.8%)	1 (0.5%)

**Table 3 medicina-57-00338-t003:** Multivariate predictors of failure.

Independent Variable	SHR	CI 95%	*p*	Coefficient	Standard Error
Age (years)	0.91	(0.86–0.96)	0.001	−0.10	0.03
Tip-Apex Distance (mm)	1.04	(0.97–1.11)	0.270	0.04	0.04
Calcar Length & Nail Length (mm)					
Short Nail	1.03	(0.98–1.07)	0.204	0.03	0.02
Long Nail	0.98	(0.93–1.04)	0.518	−0.02	0.03
Fracture Pattern & Nail Length					
2.2 Short Nail	Reference				
Long Nail	0.65	(0.01–32.24)	0.830	−0.43	1.30
2.3 Short Nail	1.05	(0.11–9.61)	0.967	0.05	1.18
Long Nail	11.14	(0.44–280.80)	0.143	2.41	18.35
3.3 Short Nail	-				
Long Nail	-				

## Data Availability

The data presented in this study are available on request from the corresponding author. The data are not publicly available due to IRB policy.

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
