# Peer review of "Medial Calcar Comminution and Intramedullary Nail Failure in Unstable Geriatric Trochanteric Hip Fractures"

_medicina, 2021, doi:10.3390/medicina57040338_

Round 1

Reviewer 1 Report

I think this is a well written article and there are little problems as investigation. And this is a very useful study in terms of the large number of cases. However, first of all, the title of the article does not match the content of the article. There is no other choice but to change it to a retrospective study of a large number of cases. There are also critical issues regarding patient selection and patient follow-up that make it difficult to accept as an article.

Author Response

Response to Reviewer 1 Comments

Point 1: I think this is a well written article and there are little problems as investigation. And this is a very useful study in terms of the large number of cases.

Response 1: We would like to thank the reviewer for their positive comments. As a high-volume hip fracture centre performing a large volume of intramedullary nails, we are in a unique position to have sufficiently powered number of cases to be analysed.

Point 2: However, first of all, the title of the article does not match the content of the article. There is no other choice but to change it to a retrospective study of a large number of cases.

Response 2: We thank you for your comment, and did not intend to mislead the reader, nor misrepresent the content of the manuscript. We have simplified the title to ‘Medial calcar comminution and intramedullary nail failure in unstable geriatric trochanteric hip fractures’.

As we only analysed variables for unstable (defined by medial calcar loss) fracture patterns, we believe that this is justified in the title. As for the retrospective natures, this is included in the abstract as the first line of methodology. We think it too cumbersome to include in the title, however based on editorial preference we are happy to do so. We agree with the reviewer that the factors towards the choice of the nail is impossible to be evaluated with this design.

 Point 3: There are also critical issues regarding patient selection and patient follow-up that make it difficult to accept as an article.

Response 3: We agree that there are issues with patient selection. Considerable bias exists for nail length, hence why we conducted the statistical analysis with interaction terms in the regression analysis. Randomised evidence has failed to recruit patients to an acceptable level to power the outcome of failure in previous publications. The study is meant to evaluate standard practice. Selection bias in practical surgery is called “surgical decision making”, which in this case probably prevents many failures, which would be apparent in a randomised design.

We have addressed many shortcomings of this study within the discussion’s ‘Limitations’ subheading. No prospective study to our knowledge has been able to follow up patients for a sufficiently long period of time.

The authors are happy to address specific concerns regarding methodology if these are rate-limiting steps towards publication. We thank reviewer 1 for their time examining this manuscript. 

Reviewer 2 Report

The study by Tarrant et al is well-designed and the results could benefit the orthopedic community. The manuscript is also well-written, and each section is adequately detailed. The interpretation of their results is also accurate. Just a few suggestions/questions before publishing-  

  1. Johnson et al (2017) recently showed that both age and activity levels were important predictors for nail failure (age was not a predictor when adjusted for ASA scores). Although this paper is referenced in your study, you do not discuss this in reference to your conclusions.
  2. Was there data available on BMI and any other cardiovascular/ metabolic comorbidities of the patients?

Minor-

  • Line 43- Stability itself has been attempted to be defined.
  • Line 57- Comma missing between ‘patterns, is’

Reviewer 3 Report

Tarrant et. al. analyzed 1,100 consecutive geriatric hip fracture patients undergoing surgical fixation using long or short version of G3 IMN nail. They found that age was associated with failure for those 16 failed cases with lateral wall insufficiency and medical calcar loss. The ms. is well written and organized. The following questions should be answered before considered for publication.

  1. Page 2, Line 83: Please provide a brief explanation on the AO/OTA subtypes AO31A2.2, 2.3 and 3.3.
  2. In this study, the median follow-up time is 707 days, while failures occurred at a median post-operative time of 111 days (40-413). Why did the failures happen during the first few months after the surgery?

Reviewer 4 Report

Overall, I think the paper is interesting..

The approach of the authors to this important topic is structured and supported with a huge numbers of cases. However - some points should be reconsidered: 

  1. Maybe the definition of failure should be reconsidered... I think a peri-implant fracture, especially after a second trauma should not be included, as this force cannot be predicted and would have an impact on any implant of the femur. However I cannot understand why the shortening of the femoral neck and thus the lateralisation of the femoral neck screw is excluded instead. I suggest that the primary outcome should be: cut-out or cut-through of the femoral neck screw, breakage of the implant and symptomatic shortening after weight bearing - as they are all due to the instability of the fracture-pattern. Non-union could be added, but it should be differentiated, whether biology or stability ist the key-factor... 
  2. It is not mentioned, whether all patients were treated with the same post-op sceme. Was there any partial weight-bearing?
  3. Patient data should include the body mass index. As only low-energy traumas were included, I would be curious to know the percentage of heavy-weight patients, especially with the A2.3 and 3.3 Fractures. May this be another factor influencing failure risk? Please discuss.. 
  4. As there are only 16 documented failures in this huge group of patients a individulized failure-analysis would be an interesting add-on. 
  5. The key-message is that younger patients have a higher risk of failure in comminuted, instable peritrochanteric fractures... How did that result influence the treatment strategy of the authors? Do you apply additional measures? Please describe/discuss..

Round 2

Reviewer 1 Report

The authors have been able to accurately respond to many of the orders from the reviewers. I have no objection to the acceptance of the paper as a retrospective study of a huge number of cases.

Reviewer 4 Report

All questions have been answered adequately and all concerns have been adressed.